# A Comprehensive Biomarker Analysis of Microsatellite Unstable/Mismatch Repair Deficient Colorectal Cancer Cohort Treated with Immunotherapy

**DOI:** 10.3390/ijms24010118

**Published:** 2022-12-21

**Authors:** Elena Élez, Núria Mulet-Margalef, Miriam Sanso, Fiorella Ruiz-Pace, Francesco M. Mancuso, Raquel Comas, Javier Ros, Guillem Argilés, Giulia Martini, Enrique Sanz-Garcia, Iosune Baraibar, Francesc Salvà, Alba Noguerido, Jose Luis Cuadra-Urteaga, Roberta Fasani, Ariadna Garcia, Jose Jimenez, Susana Aguilar, Stefania Landolfi, Javier Hernández-Losa, Irene Braña, Paolo Nuciforo, Rodrigo Dienstmann, Josep Tabernero, Ramon Salazar, Ana Vivancos

**Affiliations:** 1Colorectal Cancer Program, Medical Oncology Department, Vall d’Hebron Institute of Oncology (VHIO), 08035 Barcelona, Spain; 2Colorectal Cancer Unit, Medical Oncology Department, Catalan Institute of Oncology, L’Hospitalet de Llobregat, 08908 Barcelona, Spain; 3Cancer Genomics Group, Vall d’Hebron Institute of Oncology (VHIO), 08035 Barcelona, Spain; 4Genomics for Precision Oncology Laboratory, Fundació Institut d’Investigació Sanitària Illes Balears (IdISBa), 07120 Palma de Mallorca, Spain; 5Oncology Data Science Group, Vall d’Hebron Institute of Oncology (VHIO), 08035 Barcelona, Spain; 6Research and Development Department, Universal Diagnostics S.L., 41013 Sevilla, Spain; 7Departament of Precision Medicine, Università degli Studi della Campania Luigi Vanvitelli, 81100 Naples, Italy; 8Medical Oncology, IOB—Hospital Quirón, 08023 Barcelona, Spain; 9Molecular Oncology Group, Vall d’Hebron Institute of Oncology (VHIO), 08035 Barcelona, Spain; 10Molecular Prescreening Program, Vall d’Hebron Institute of Oncology (VHIO), 08035 Barcelona, Spain; 11Department of Pathology, Vall d’Hebron University Hospital, 08035 Barcelona, Spain; 12Medical Oncology Department, Research Unit for Molecular Therapy of Cancer, Vall d’Hebron Institute of Oncology (VHIO), 08035 Barcelona, Spain; 13Medical Oncology Department, Catalan Institute of Oncology, Oncobell Program (IDIBELL), CIBERONC, L’Hospitalet de Llobregat, 08908 Barcelona, Spain

**Keywords:** MSI-H/dMMR, colorectal cancer, immunotherapy, biomarkers

## Abstract

The search for immunotherapy biomarkers in Microsatellite Instability High/Deficient Mismatch Repair system (MSI-H/dMMR) metastatic colorectal cancer (mCRC) is an unmet need. Sixteen patients with mCRC and MSI-H/dMMR (determined by either immunohistochemistry or polymerase chain reaction) treated with PD-1/PD-L1 inhibitors at our institution were included. According to whether the progression-free survival with PD-1/PD-L1 inhibitors was longer than 6 months or shorter, patients were clustered into the IT-responder group (*n*: 9 patients) or IT-resistant group (*n*: 7 patients), respectively. In order to evaluate determinants of benefit with PD-1/PD-L1 inhibitors, we performed multimodal analysis including genomics (through NGS panel tumour-only with 431 genes) and the immune microenvironment (using CD3, CD8, FOXP3 and PD-L1 antibodies). The following mutations were more frequent in IT-resistant compared with IT-responder groups: *B2M* (4/7 versus 2/9), *CTNNB1* (2/7 versus 0/9), and biallelic *PTEN* (3/7 versus 1/9). Biallelic *ARID1A* mutations were found exclusively in the IT-responder group (4/9 patients). Tumour mutational burden did not correlate with immunotherapy benefit, neither the rate of *indels* in homopolymeric regions. Of note, biallelic *ARID1A* mutated tumours had the highest immune infiltration and PD-L1 scores, contrary to tumours with *CTNNB1* mutation. Immune microenvironment analysis showed higher densities of different T cell subpopulations and PD-L1 expression in IT-responders. Misdiagnosis of MSI-H/dMMR inferred by discordances between immunohistochemistry and polymerase chain reaction was only found in the IT-resistant population (3/7 patients). Biallelic *ARID1A* mutations and Wnt signalling activation through *CTNNB1* mutation were associated with high and low T cell immune infiltrates, respectively, and deserve special attention as determinants of response to PD-1/PD-L1 inhibitors. The non-MSI-H phenotype in dMMR is associated with poor benefit to immunotherapy. Our results suggest that mechanisms of resistance to immunotherapy are multi-factorial.

## 1. Introduction

Immunotherapy has been incorporated in the treatment of metastatic colorectal cancer (mCRC) with the Microsatellite Instability High or Deficient Mismatch Repair system (MSI-H/dMMR) phenotype. Pembrolizumab and nivolumab (single agent or combined with ipilimumab) have demonstrated long-lasting responses ranging from 31% in pre-treated patients to 60% in the front-line setting [1,2,3,4,5,6]. 

Molecular biology surrounding MSI-H/dMMR CRC favours the response to PD-1 and PD-L1 inhibitors (anti-PD-1/PD-L1). A dysfunctional MMR system leaves unrepaired DNA alterations, mainly insertions and deletions in codons that modify the reading frame, resulting in Tumour Mutational Burden (TMB) over 10 mutations per megabase (Mb) with high immunogenic potential [7,8,9]. Further genomics, MSI-H/dMMR CRC is mainly clustered in Consensus Molecular Subtype 1 [10] and associates immune infiltrates composed of different T cell subpopulations and cells belonging to innate immunity [11]. However, this phenotype also exemplifies adaptive immune resistance through different mechanisms. High expression of immune checkpoint proteins [12,13], mutations in genes related to antigen processing and presentation machinery [14] or interferon signalling [15] are some examples. 

Since 30–50% of MSI-H/dMMR mCRC do not benefit from anti-PD-1/PD-L1 [2,4,5], a better understanding of the molecular traits associated with these drug outcomes is an unmet need. Lynch syndrome diagnosis, PD-L1 expression and presence of *BRAF*/*RAS* mutations were explored in trials, but do not clearly correlate with response to anti-PD-1/PD-L1 [3,4,5]. Other preliminary data showed *B2M* and *JAK1* mutations as mechanisms of primary resistance to anti-PD-1/PD-L1, as well as secondary resistance, but only for *B2M* mutations [15,16,17]. Regarding tumour microenvironment, PD-L1 expression at invasive front combined with higher presence of extracellular mucin was related with higher clinical activity to PD-1 blockade, although further confirmatory analysis is warranted [18]. 

Here, we present a single institution cohort of MSI-H/dMMR mCRC patients treated with anti-PD-1/PD-L1. The aim of the study is to perform a comprehensive tumour characterisation in terms of genomics, immune microenvironment and MSI-H/dMMR diagnosis techniques to decipher patterns of benefit/resistance to anti-PD-1/PD-L1. 

## 2. Results

### 2.1. Cohort Description

The study included 16 patients (Table 1). Median age at diagnosis was 55 years (range, 30–90 years). Females represented 56.3% of the population. Right-sided tumours prevailed (56.2%), and 62.4% of patients had metastases in at least two organs. The median follow-up was 56.4 months (min-max: 22.2–87.8). Median progression free survival (mPFS) to anti-PD-1/PD-L1 was 14.5 months (95% CI, 2.1-NR), and median overall survival (mOS) of metastatic disease was 63.9 months (95% CI, 45.6-NR) (Appendix A). Individual data are shown in Figure 1.

### 2.2. Cohort Genomic Profile

The mutational landscape of the cohort, according to the NGS analysis, is summarised in Figure 2 (see also Appendix A). Of note, patients harbouring mutations in *CTNNB1*, *EGFR* and *B2M* genes were higher in IT-resistant compared with IT-responders (Figure 2A). Regarding biallelic mutations (Figure 2B), those found in the *APC* tumour suppressor gene and in MMR genes were related to the underlying mechanisms of carcinogenesis. *MLH1*, *MSH2* and *MSH6* were mutated in both alleles in one patient each, corresponding to 3 out of the 4 Lynch syndrome cases, as expected (Figure 3). 

*PTEN* and *ARID1A* truncating mutations were identified in both groups, with frequencies of 31.2% and 50%, respectively (Figure 2A). In order to help give a context to our results, we analysed publicly available genomic data from 1134 colorectal tumours with known MSI/MMR status [19] through CBioportal [20]. The overall mutation rate was 32.4% in *PTEN* and 55.2% in *ARID1A* in 105 MSI-H/dMMR CRC samples analysed through the MSK-IMPACT platform [19]. Hence, our cohort is comparable in mutation prevalence to that of a larger cohort, suggesting appropriate diversity in spite of the limited sample size. When considering biallelic mutations, the rate in our cohort for both *PTEN* and *ARID1A* was 25% (Figure 2B), and 13.3% and 14.3%, respectively, according to cBioPortal data [20]. Interestingly, in our cohort, biallelic *ARID1A* mutations were exclusively found in IT-responder, present in 44% of patients (4 out of 9 in IT-responder *versus* none in IT-resistant). On the contrary, biallelic *PTEN* mutations were predominant among IT-resistant and present in 43% of the cases versus 11% of the cases from the IT-responder (3 out of 7 versus 1 out of 9) (Figure 2B). 

### 2.3. TMB Scores and Indels

The median values of TMB were 47.2 mut/Mb (range 26.6–70.0) and 38.8 mut/Mb (range 8.4–60.1) in IT-resistant and IT-responder, respectively, and there was no association between TMB and anti-PD-1/PD-L1 benefit (Appendix A). Tumours from patients 10 and 14 harboured the highest TMB (Figure 3, Appendix A), but they were clustered in the IT-resistant group and showed short stable disease and progressive disease as the best response to anti-PD-1/PD-L1, respectively (Figure 1). Interestingly, patient 10 presented with a functional mutation in *CTNNB1* as well as a biallelic mutation in *PTEN* (Figure 3). Patient 14 should be interpreted with caution since TMB could have been overestimated because the sample used for analysis had been exposed to previous chemotherapy [21] (Appendix A). No remarkable differences were observed between IT-resistant and IT-responder according to the *indel* rate (Figure 3, Appendix A). According to COSMIC signature analysis [22], the whole cohort had a strong component of DNA mismatch repair deficiency with small *indels*, without relevant differences in terms of the relative proportion of this signature compared with others between IT-responder and IT-resistant (Figure 3, Appendix A). 

### 2.4. Immune Microenvironment

Densities of all lymphocyte’s subpopulations were higher in IT-responder, as well as PD-L1 CPS (Figure 3). The median densities in IT-responder compared with IT-resistant were 885 (range 507–1517) versus 541 (range 282–925), 459 (range 334–1081) versus 242 (range 110–496) and 69 (range 34–142) versus 16 (range 9–42) for CD3^+^, CD8^+^ and FOXP3^+^ cells, respectively. These differences were not statistically significant, except for FOXP3^+^ cells (*p* = 0.01). (Appendix A). The median PD-L1 CPS was also higher in IT-responder (7.5, range 1–90) compared with IT-resistant (3.5, range 1–5), but not statistically significant (Appendix A).

It is noteworthy that tumours harbouring biallelic mutation in *ARID1A* showed the highest levels of CD3^+^ cells density (patients 2, 9 and 7), as well as of CD8^+^ (patients 2 and 7), and FOXP3^+^ (patients 2 and 7) (Figure 3).

Of note, tumour from patient 10 with biallelic mutation in *PTEN* and activating mutation in Wnt pathway, *CTNNB1*, presented the lowest density of CD3^+^ and CD8^+^ cells of the entire cohort (Figure 3). 

### 2.5. Diagnosis Techniques of MSI-H/dMMR Phenotype 

Central IHC results were available for 15 out of 16 patients (Appendix A), and all were compatible with dMMR. Regarding the PCR technique, central results were available for 15 out of 16 patients, but they were compatible with MSS in 3 out of 7 patients in the IT-resistant group and discordant with their paired IHC analysis (Figure 3). 

## 3. Discussion

We show a molecular analysis of 16 MSI-H/dMMR mCRC patients treated with anti-PD-1/PD-L1 aiming to provide new insights into the immunotherapy biomarkers of response and resistance. No differences were observed between groups in clinical terms or according to *RAS*/*BRAF* mutational status, but it was remarkable that responses to anti-PD-1/PD-L1 in the first-line setting lasted for at least 3 years. Genomic analysis identified an enrichment of mutations in *B2M* and *CTNNB1* genes in IT-resistant group. Biallelic and monoallelic mutations in *B2M* have been previously associated in single patients with primary [16] and secondary [17] resistance to anti-PD-1/PD-L1 in MSI-H/dMMR CRC, respectively, although a larger series has shown the opposite [23]. Regarding the *CTNNB1* gene, Wnt pathway activation is associated with a cold immunophenotype potentially linked to a reduction of T cell infiltration through failure in dendritic cell recruitment [24]. Of note, patient 10 harbouring the *CTNNB1* mutation also showed the lowest T cell infiltration, which could justify the absence of anti-PD-1/PD-L1 benefit in spite of having the highest TMB. Mutation p.Gly659fs in the *RNF43* gene, a hotspot mutation in MSI-H/dMMR tumours and component of the Wnt pathway, was broadly distributed in the cohort, but recent data have described no functional significance of this mutation in terms of pathway activation [25]. When we focused on biallelic mutations, *ARID1A* was only found in IT-responders and, on the contrary, *PTEN* was enriched in IT-resistant. According to the rate of biallelic mutations in these genes surveyed in cBioPortal [20], our results raise the possibility of potential anti-PD-1/PD-L1 biomarkers in a substantial proportion of MSI-H/dMMR patients (27%) if they are further confirmed. *ARID1A* belongs to the chromatin remodelling complex SWItch/Sucrose Non-Fermentable (SWI/SNF). A protein-protein interaction between ARID1A and MSH2 exists, as ARID1A depletion results in a lower functional capacity of the MMR system [26]. Since biallelic mutations would lead to the absence of ARID1A protein expression, the higher impairment on the MMR system in terms of DNA repair would generate a higher rate of neoantigens that could justify higher benefit to anti-PD-1/PD-L1. Furthermore, tumours harbouring biallelic *ARID1A* mutations showed high levels of PD-L1 expression and high densities of T cell subpopulations. Previous reports have also described such immune contexture for *ARID1A* mutated tumours without specifying the heterozygosity [27,28], but whether this condition could be a biomarker of immunotherapy benefit in solid tumours is still debated in the literature [29,30]. *PTEN* loss promotes an immunosuppressive context through releasing cytokines and chemokines that boost the proliferation and differentiation of myeloid-derived suppressor cells and M2 macrophages, the expansion of regulatory T cells and the reduction of Natural Killer cells [31,32,33]. In fact, *PTEN* mutations have been associated with primary and secondary immune resistance in other malignancies [33], including MSI-H/dMMR gastrointestinal tumours [34]. We hypothesised that complete loss of *PTEN,* regardless of *PIK3CA* mutations, could mediate immune resistance in our cohort through different mechanisms that deserve further investigation, including deep tumour microenvironment analysis. TMB did not correlate with anti-PD-1/PD-L1 benefit in our cohort, contrary to other reports. TMB analysis through the Foundation Medicine platform of a cohort of 22 patients with MSI-H/dMMR CRC treated with immunotherapy pointed out a cut-off between 37–41 mut/Mb to discriminate between responders and non-responders (SD or PD) [35]. The authors also concluded that a TMB of 61.8 mut/Mb was the 75th percentile for MSI-H/dMMR CRC, after surveying more than 800 tumours with this phenotype in the Foundation Medicine Database. Of note, TMB from patients 10 and 14 were above 61.8 mut/Mb, but they belong to IT-resistant group. This result points out the limitations of TMB as a single predictive biomarker of immunotherapy response, mainly because of the lack of a standard cut-off and because it does not contribute to a comprehensive evaluation of mutational immunogenic potential [36,37]. Regarding the immune microenvironment, different T cell subpopulation densities and PD-L1 expression were clearly higher in IT-responder, with no direct correlation with TMB, *indel* ratio, or small *indel* COSMIC signature. Despite the limited availability of samples preventing us from performing immune characterisation in the whole cohort, as well as spatial resolution and functional analysis, our data suggest a potential stratification of cold and hot MSI-H/dMMR tumours to better predict immunotherapy benefits. However, further evaluation is needed. Finally, misdiagnosis of MSI-H/dMMR phenotype due to discordances between IHC and PCR techniques were only found in IT-resistant. These results are aligned with those from Cohen et al., who postulated misdiagnosis as a potential cause of primary resistance to immunotherapy resistance in MSI-H/dMMR mCRC [38]. Similarly, a phase II trial with avelumab in patients with MSI-H/dMMR mCRC showed a higher response rate when both PCR and IHC concluded a deficiency in the MMR system [39]. As a consequence, we support ESMO recommendations in terms of using both PCR and IHC to evaluate the MMR system when immunotherapy is planned [40]. However, we must consider that PCR and IHC have technical limitations and that patient samples are not always as optimal as they should be for analysis. For this reason, we encourage not excluding the use of immunotherapy in patients with misdiagnosis since 14 out of 74 patients included in the Checkmate 142 clinical trial locally categorised as MSI-H/dMMR either by IHC or PCR were centrally reclassified as MSS by PCR, although this change did not seem to impact the benefit of immunotherapy [4]. 

The main limitation of our work is the limited number of patients, due to the low incidence of MSI-H/dMMR among mCRC. The inclusion of patients who received different schemas of treatment also represented a lack of homogeneity in terms of the mechanisms of action of immunotherapy, although all of them included an anti-PD-1 or anti-PD-L1 agent. 

Overall, our results indicate that MSI-H/dMMR mCRC is a molecular heterogeneous disease and upfront determinants of resistance to anti-PD-1/PD-L1 are likely multifactorial. The development of immunotherapy biomarkers should rely on combined tumour genomic profiling (beyond TMB) and its leading immune-contexture. A recent analysis showed a higher response rate to pembrolizumab in tumours with high TMB and T cell-inflamed gene expression profiles [41] compared with those with only one of these features. Further studies are required to confirm our findings, but biallelic *ARID1A* mutations and Wnt signalling activation through *CTNNB1* mutation are associated with high and low T cell immune infiltrates, respectively, and deserve special interest. Regarding MSI-H/dMMR diagnosis phenotype, analysis by IHC, PCR and NGS, if feasible, should be performed if clinically suspected to ensure optimal identification of potential mCRC candidates for immunotherapy, although misdiagnosis among techniques could lead to an absence of benefit to immunotherapy.

## 4. Materials and Methods

### 4.1. Patients and Samples

Between January 2017 and May 2019, patients who met the following inclusion criteria were prospectively selected for the study cohort at Vall d’Hebron University Hospital: 1. Diagnosis of MSI-H/dMMR mCRC by either immunohistochemistry (IHC) or polymerase chain reaction (PCR) techniques. 2. Treatment with anti-PD-1/PD-L1 in a clinical trial setting or as per standard practice in immunotherapy naïve context. 3. Availability of formalin-fixed embedded paraffin (FFPE) samples for genomic analysis. Of the 19 patients identified, 16 were finally included and 3 were excluded because of sample failure. Regarding anti-PD-1/PD-L1 therapy, all but one patient was enrolled in clinical trials testing the regimens specified in Table 1 until progression and/or unacceptable toxicity. The remaining patient was treated with an anti-PD-1 drug until progression. 

Beyond genomics, the remaining analysis depended on the availability of tumour material because no patient biopsies were planned only for investigational purposes (Appendix A). The study was approved by the Vall d’Hebron University Hospital institutional ethical review board.

### 4.2. Library Preparation and Bioinformatic Analysis

Genomic analysis was performed using a hybrid capture-based custom Next-Generation Sequencing (NGS) panel designed for the analysis of FFPE-derived genomic DNA (Appendix A). *Library preparation* and *Bioinformatic analysis* are detailed in Appendix A). The test captures the exonic regions of 431 relevant genes for pan-cancer study, as well as a backbone of single nucleotide polymorphisms (SNPs) spread along the whole genome and with an average population frequency of 0.5. Copy number alterations (CNA) and loss of heterozygosity (LOH) were also analysed (see Appendix A). Mutational signatures were defined according to version 3.1. June 2020 of the Catalog of Somatic Mutations in Cancer (COSMIC), referring to single-base substitutions, considers the mutated base, as well as the bases immediately 5′ and 3′ [22].

TMB was calculated as the number of non-synonymous variants (filtering out the SNPs) per Mb. The rate of insertions and deletions (*indels*) of the total mutation pool was also determined. 

### 4.3. Immune Microenvironment Analysis

IHC was performed on available FFPE tumour sections using CD3, CD8, FOXP3 and PD-L1 antibodies. Densities of lymphocytes CD3^+^, CD8^+^ and FOXP3^+^ were quantified through digital image analysis. For PD-L1 determination, the Composite Positive Score (CPS) was calculated dividing the number of PD-L1 positive cells by the total number of Pan-Keratin positive tumour cells multiplied by 100 (detailed procedures and codes, clones and providers for the antibodies used in the study are listed in Appendix A. *Immune microenvironment analysis*).

### 4.4. MSI/MMR Status Analysis

Central revision of MSI-H/dMMR phenotype by either IHC or PCR was planned in each case. MSI-H/dMMR phenotype diagnosis followed ESMO recommendations: loss of nuclear expression of at least 1 out of 4 proteins in IHC test performed using regular MLH1, MSH6, PMS2, MSH2 antibodies, or instability in at least 2 out of 5 microsatellites in PCR analysis performed through a five microsatellite panel (BAT25, BAT26, NR21, NR24 and MONO27) [40]. The commercial kit Promega MSI Analysis System version 1.2 was used for this purpose.

### 4.5. Statistical Analysis

According to individual Progression Free Survival (PFS) anti-PD-1/PD-L1 patients were clustered into two groups: *IT-resistant* when PFS was up to 6 months, or *IT-responder* when it was longer. PFS was defined as the period of time between the date of anti-PD-1/PD-L1 initiation and the date at which progression was documented according to RECIST v1.1 [42] or the last follow-up date (censoring), whichever occurred first. Pseudoprogression was ruled out due to the clinical course of the patients and the low chance of facing this situation in MSI-H/dMMR CRC [43]. Metastatic OS (OS) was defined as the period of time between the date of metastatic disease diagnosis and the date at which cancer death was documented or the last date of follow-up (censoring), whichever occurred first. All analyses were carried out using R version 4.0.3 statistical software. The median follow-up was calculated using the Kaplan–Meier reverse method. Descriptive and survival analyses are detailed in the Appendix A.

The cBioPortal for Cancer Genomics database was planned to be surveyed in order to check the prevalence of the most relevant genomic alterations observed in genomic analysis [20]. 

## 5. Conclusions

-Biallelic *ARID1A* mutation and *CTNNB1* mutation are associated with high and low T cell immune infiltrates, respectively, in MSI-H/dMMR CRC.-Both *ARID1A* and *PTEN* biallelic mutations should be further investigated as immunotherapy biomarkers in MSI-H/dMMR CRC.-TMB does not correlate with immunotherapy benefits based on PD-1/PD-L1 inhibitors in MSI-H/dMMR CRC.-Discordances in MSI-H/dMMR assessment between IHC and PCR are associated with limited benefit to immunotherapy.-Immunotherapy biomarkers in MSI-H/dMMR mCRC should rely on tumour genomic profiles and their dominant immune-contexture.

## Figures and Tables

**Figure 1 ijms-24-00118-f001:**
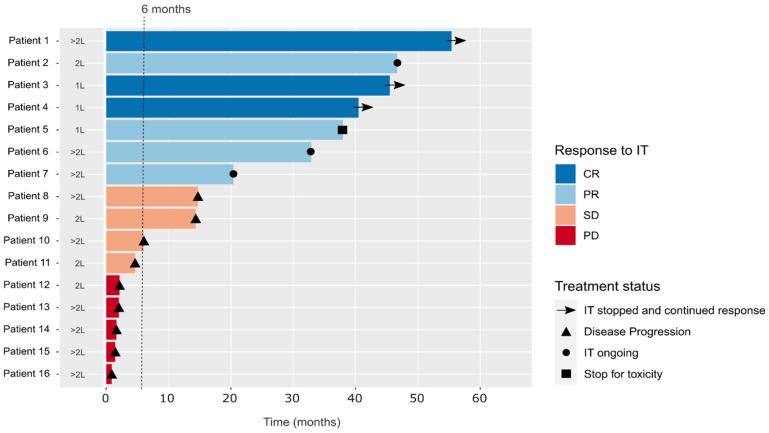
Line of anti-PD-1/PD-L1 administration (1L: first line, 2L: second line, >2L: 3rd line and beyond), PFS to anti-PD-1/PD-L1 for individual patients, best response to anti-PD-1/PD-L1 according to RECIST v1.1. (CR: complete response, PD: progression disease, PR: partial response, SD: stable disease) and treatment status at the last follow-up. None of the patients who presented CR or PR progressed, even if anti-PD-1/PD-L1 was stopped per protocol or because of toxicity. All patients with SD as the best response finally presented with PD. The dotted line shows the cut-off of 6 months for PFS that divide IT-respond than IT-resist. IT: immunotherapy referred to anti-PD-1/PD-L1.

**Figure 2 ijms-24-00118-f002:**
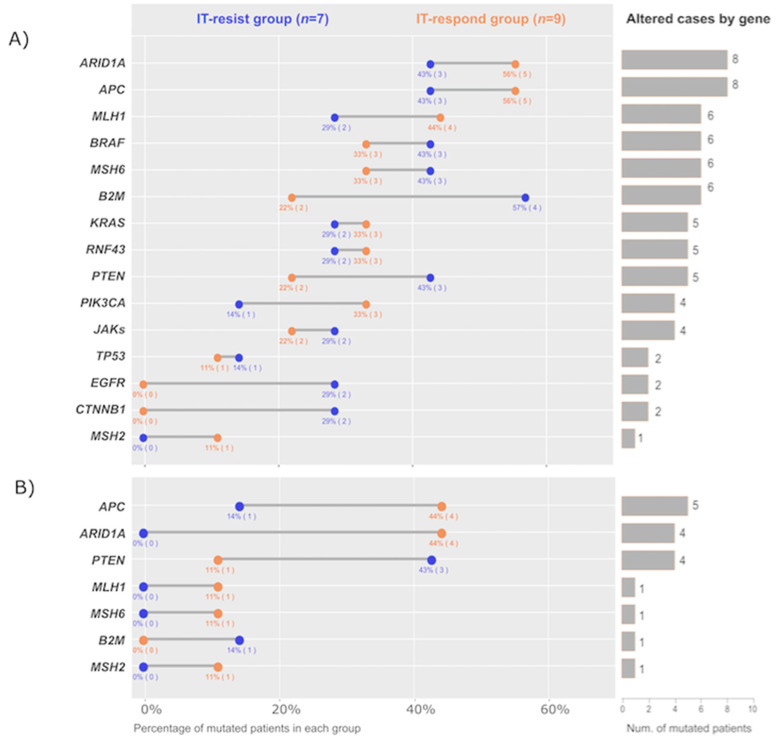
Mutational landscape of the cohort. Genes belong to CRC carcinogenesis-related pathways, such as Wnt (*APC*, *RNF43* p.Gly659fs, *CTNNB1*), MAPK (*BRAF*, *KRAS*, *EGFR*) or PI3K (*PTEN*, *PIK3CA*), to antigen processing and presentation machinery (*B2M*), to MMR system (*MLH1*, *MSH6*, *MSH2*), to interferon signalling (*JAK* 1, 2 or 3), or to master genetic or epigenetic regulators, such as *TP53* or *ARID1A*, respectively. The right column shows the number of patients harbouring a given gene mutation. The percentage of patients harbouring a given gene mutation and the absolute number of patients in brackets are represented in blue and orange for the IT-resistant group and the IT-responder group, respectively, at each extrem of the bars. (**A**) refers to any mutation. (**B**) refers to biallelic mutations.

**Figure 3 ijms-24-00118-f003:**
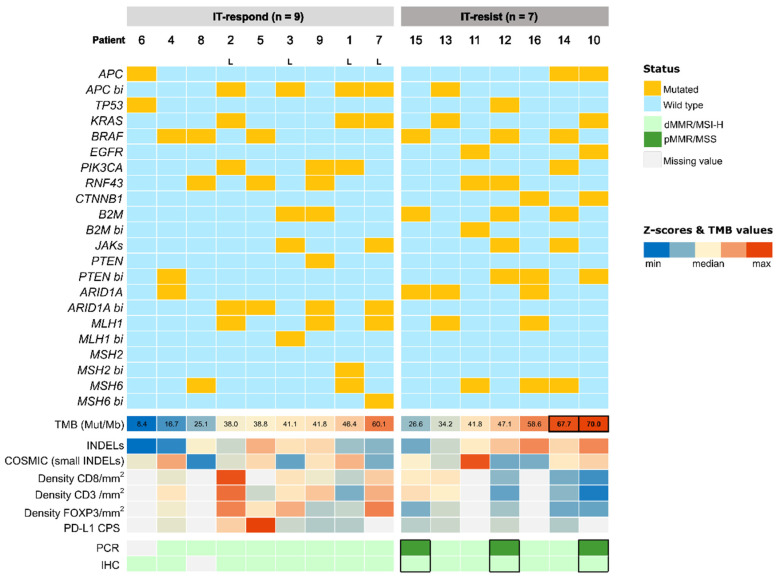
Representative heatmap of the entire cohort, including genomics (mutational profile—monoallelic and biallelic mutations are shown separately for a given gene, TMB, indels ratio and small indels COSMIC signature), immune microenvironment (CD3^+^, CD8^+^ and FOXP3^+^ densities and PD-L1 CPS), and MSI/MMR status evaluation through IHC and PCR, according to IT-responder and IT-resistant. Bi: biallelic. L: Lynch syndrome.

**Table 1 ijms-24-00118-t001:** Cohort description. Characteristics are specified for the overall population and for each group (IT-responder and IT-resistant). Anti-PD-1/PD-L1 response is defined according to RECIST v1.1. The percentage of patients diagnosed with Lynch syndrome in a genetic counselling unit and/or receiving anti-PD-1/PD-L1 in 1st line setting are in italics since they are all clustered in IT-responder. Mutations in *KRAS* and *BRAF* V600E determined by standard practice were similarly distributed between the groups. CR: complete response. PD: progression disease. PR: partial response. SD: stable disease.

	Number of Patients *n* = 16	%	IT-Respond *n* = 9	%	IT-Resist *n* = 7	%
**Median age at diagnosis and range**	55.4	(30–90)	57.22	(31–90)	53.14	(30–71)
**Gender**						
Male	7	43.7%	4	44.4%	3	42.8%
Female	9	56.3%	5	55.5%	4	57.1%
**Stage at diagnosis**						
II	1	6.2%	1	11.1%	0	0%
III	8	50%	3	33.3%	5	71.4%
IV	7	43.7%	5	55.5%	2	28.6%
**Primary tumor location**						
Right	9	56.2%	5	55.5%	4	57.1%
Transverse	3	18.7%	1	11.1%	2	28.6%
Left-rectum	4	25%	3	33.3%	1	14.3%
**Number of metastatic sites**						
1	6	37.5%	4	44.4%	2	28.6%
2	7	43.7%	4	44.4%	3	42.8%
3	3	18.7%	1	11.1%	2	28.6%
**Anti-PD-1/PD-L1 setting**						
1st line	3	*18.7%*	3	33.3%	0	0%
2nd line	4	25%	2	22.2%	2	28.6%
>2nd line	9	56.2%	4	44.4%	5	71.42%
** *KRAS* ** **mutation**						
Yes	5	31.2%	3	33.3%	2	28.6%
No	11	68.7%	6	66.6%	5	71.4%
** *NRAS* ** **mutation**						
Yes	0	0%	0	0%	0	0%
No	0	0%	0	0%	0	0%
** *BRAF* ** **V600E**						
Yes	6	37.5%	3	33.3%	3	42.8%
No	10	62.5%	6	66.6%	4	57.1%
**Lynch syndrome**						
Yes	4	*25%*	4	44.4%	0	0%
No	12	75%	5	55.5%	7	100%
**Anti-PD-1/PD-L1 regimen**						
Anti-PD1	7	43.7%	5	55.5%	2	28.6%
Anti-PD-L1	3	18.7%	1	11.1%	2	28.6%
**Anti-PD-L1 combo:**	6	37.5%	3	33.3%	3	42.8%
*+Bevacizumab*	3	18.7%	2	22.2%	1	14.3%
*+CD40 agonist*	1	6.2%	0	0%	1	14.3%
*+anti-CEA-CD3 antibody*	2	12.5%	1	11.1%	1	14.3%
**Anti-PD-1/PD-L1 response**						
CR	3	18.7%	3	33.3%	0	0%
PR	4	25%	4	44.4%	0	0%
SD	4	25%	2	22.2%	2	28.6%
PD	5	31.2%	0	0%	5	71.4%

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
