# Peer review of "A Comprehensive Biomarker Analysis of Microsatellite Unstable/Mismatch Repair Deficient Colorectal Cancer Cohort Treated with Immunotherapy"

_ijms, 2022, doi:10.3390/ijms24010118_

Round 1

Reviewer 1 Report

ARID1A mutation and CTNNB1 mutation correlation with T cells infiltration  which is very useful for cancer therapy.

It would be nice if they show any cancer cell with T cell coculture with ARID1A and PTEN mutation and T cell function.

Author Response

Dear Reviewer,

Thank you very much for your suggestion. Even though it would be a very informative experiment, we are not able to perform this analysis since the project was exclusively designed for being developed using human tumor samples. However, we will consider it for further research.

Reviewer 2 Report

The authors present here a well-written and documented work. I appreciate it very much and I consider this a valuable work worthy to be published. The major weakness of the study is the low number of patients enrolled. In addition, I have some minor improvement recommendations:

- the therapeutic protocol of the patients should be clearly stated in section 2.1 (received therapy, duration, etc)

- please provide the codes, clones, and provider of the antibodies used

- please provide the primers used for PCR in a table

- please provide more information regarding the NGS analysis (NGS platform used, extraction protocol and kit, etc)

Author Response

Dear Reviewer,

Thank you very much for your feedback and for your recommendations. Please, find below the commentaries related with them.

Regarding anti-PD-1/PD-L1 therapy, all but one patient were treated in clinical trials testing the regimens specified in Table 1, until progression and / or unacceptable toxicity. The remaining patient was treated with an anti-PD-1 drug until progression. Main text has been amended (please, see lines 93 -96).

- please provide the codes, clones, and provider of the antibodies used

The codes, clones and providers for the antibodies used in the study are listed in “Supplementary material. Immune microenvironment analysis”. Main text has also been amended (please, see lines 120-122)

- please provide the primers used for PCR in a table

Unfortunately, we can’t provide this information because we used the commercial kit Promega MSI Analysis System version 1.2, and it is not available in the datasheet. Main text has been amended (please, see lines 129-130)

- please provide more information regarding the NGS analysis (NGS platform used, extraction protocol and kit, etc)

Technical information regarding NGS analysis is provided in “Supplementary material. Library preparation and Bioinformatic analysis”. Main text has also been amended (please, see lines 104-105).

Reviewer 3 Report

In this manuscript, Elena Elez and collaborators performed multimodal analysis including genomics (through NGS panel tumour-only with 431 genes) and immune microenvironment (using CD3, CD8, FOXP3 and PD-L1 antibodies) in sixteen patients with mCRC and MSI-H/dMMR. The patients were clustered into IT- responder group or IT- resistant group, according to whether the progression free survival with PD-1/PD-L1 inhibitors was longer than 6 months or shorter. They found that B2M, CTNNB1, and biallelic PTEN mutations were more frequent in IT-resistant compared with IT-responder groups. In contrast, biallelic ARID1A mutations were exclusively found in IT-responder. In addition, biallelic ARID1A mutations and Wnt signalling activation through CTNNB1 mutation associate with high and low T cell immune infiltrates.

The manuscript is of interest. Although the number of mCRC patients analysed in this study is limited, the results indicate that tumour mutational burden did not correlate with immunotherapy benefit suggesting that the mechanisms of resistance to anti-PD-1/anti PD-L1 are multi-factorial.

Questions:

Did the authors use macrophages and myeloid-derived suppressor cell biomarkers in the immune microenvironment analysis? Is there any correlation with the resistance to anti-PD1/PD-L1 immunotherapy?

Author Response

Dear Reviewer,

Thank you very much for your suggestion. Unfortunately, we did not analyse these immune cells subtypes and, therefore, we did not assess the potential correlation with anti-PD-1/PDL-1 immunotherapy outcomes. The proposal is really interesting, however, we did not consider it before neither at the present time because of exhaustion of  tumour sample in the vast majority of patients.

Round 2

Reviewer 1 Report

I really appreciate the author's explanation.